# Accelerated Resolution Therapy (ART) for the treatment of posttraumatic stress disorder in adults: A systematic review

David Paul Storey[1,2☯]*, Emily Claire Shaw Marriott[1,2☯], Joshua A. Rash[1,2,3]

**1** Department of Psychology, Memorial University of Newfoundland, St. John's, NL, Canada, **2** Memorial University of Newfoundland Behavioural Medicine Centre (MUN-BMC), Memorial University of Newfoundland, St. John's, NL, Canada, **3** Centre for Health Policy and Inequalities Research (CHPIR), Duke University, Durham, NC, United States of America

☯ These authors contributed equally to this work.
* dpstorey@mun.ca

**Data Availability Statement:** Data and related metadata supporting the findings in this review can be found in the OSF repository: Storey DP, Marriott ECS, Rash J. Accelerated Resolution Therapy

## Abstract

Accelerated Resolution Therapy (ART), developed in 2008, uses techniques such as rapid eye movement, in vivo exposure, and image rescripting to recondition stressful memories, and reduce physical and emotional reactions to traumatic memories. There is considerable interest in evidence-based treatments for post-traumatic stress disorder (PTSD). This is the first systematic review examining the efficacy of ART for the treatment of PTSD among adults. We searched MEDLINE, PsycINFO, Embase, CINAHL, Scopus, trial registries, and government and private websites for citations published before October 2023. Studies that reported on the effect of ART for PTSD among adults were included. Meta-analyses could not be undertaken due to heterogeneity in study designs and an insufficient number of studies with a low risk of bias. Risk of bias was assessed, and findings synthesized following the synthesis without meta-analysis (SWiM) guidelines. Of the 112 records screened, five studies ($N_{enrolled}$ = 337; $N_{completed}$ = 250) and six reports of studies met criteria for inclusion. Included studies reported a significant reduction in symptoms of PTSD from pre- to post-intervention, $d$ = 1.12 to 3.28. Significant reductions were also reported in symptoms of depression, mental distress, anxiety, and sleep dysfunction. ART shows some promise as a time-efficient clinical treatment for symptoms of PTSD in adults; however, more high-quality studies are needed.

## Introduction

While estimates vary, posttraumatic stress disorder (PTSD) is a prevalent and debilitating disorder. In a systematic review of the literature on PTSD in primary care settings globally ($k$ = 27, $N$ = 24,998), Greene, Neria [1] reported that prevalence rates ranged between 2% and 39% with significant heterogeneity in estimates explained by the level of trauma exposure among samples. In a systematic review of 27 studies ($N$ = 30,878) reporting on the prevalence of mental health conditions among ambulance personnel, a population regularly exposed to

(ART) for the treatment of posttraumatic stress disorder in adults: A systematic review [Internet]. OSF; 2024. Available from: osf.io/9wh4q. **Data Repository** Additional information not included in the main text or supplementary files can be accessed at the following link: https://osf.io/9wh4q/ . Information provided here includes: a) table of all data extracted from the primary sources for the review; b) responses to requests for information from authors; c) forest plot calculations; d) effect size calculations for Kip et al., 2015; standard deviation calculations for Rossiter et al., 2017; and e) a copy of Witt's 2019 dissertation.

**Funding:** The authors received no specific funding for this work.

**Competing interests:** The authors have declared that no competing interests exist.

trauma, Petrie, Milligan-Saville [2] estimated the prevalence rate of PTSD at 11% worldwide. According to the U.S. Department of Veterans Affairs National Center for PTSD [3] about 6% of the U.S. general population will have PTSD at *some* point in their lives and in any given year about 5% of the U.S. general population has active PTSD symptoms. In the general population PTSD is more common in women (8%) than men (4%), largely due to women's increased exposure to sexual violence [3]. In military and veteran populations rates of PTSD are higher for both men and women. Mota, Tsai [4] examined the prevalence and clinical correlates of subthreshold PTSD on a sample of 1,484 U.S. military veterans who participated in the National Health and Resilience in Veterans Study (NHRVS). They observed that approximately one in three U.S. veterans experience clinically significant PTSD symptoms in their lifetime, and that subthreshold PTSD is associated with an elevated burden of comorbid psychiatric disorders, such as depression and anxiety. The presence of PTSD or posttraumatic stress symptoms (PTSS) is associated with greater frequency and severity of pain, cardio-respiratory symptoms, and gastrointestinal symptoms, as well as poor health related quality of life and poor work and family functioning [5, 6].

There are three interventions strongly recommended for the treatment of PTSD: trauma focused cognitive behavioural therapy (CBT), cognitive processing therapy (CPT), and prolonged exposure (PE) [7]. Trauma focused CBT is a 12–16 session present focused treatment that aims to understand how a client's current thoughts, feelings, and behaviors around a trauma are maintaining their difficulties, and then works with the client to help them get 'unstuck' by applying principles from both cognitive and behavioral psychology (e.g., challenging unhelpful beliefs, activity scheduling, etc.). CPT for PTSD is a 12-session treatment which focuses initially on why a trauma occurred and then emphasizes identifying, challenging, and then modifying unhelpful beliefs related to the trauma through the use of progressive worksheets. PE for PTSD is an 8–15 session treatment consisting of imaginal exposure—recounting the traumatic memory and processing the experience—and in vivo exposure—repeatedly confronting trauma-related stimuli that are now safe but were previously avoided. Additional interventions that have some support are eye movement desensitization and reprocessing (EMDR) therapy and pharmacotherapy, such as sertraline, paroxetine, fluoxetine, and venlafaxine [8]. EMDR is a 6–12 session exposure-based therapy that pairs rapid eye movements—which proponents claim facilitate processing and integration—with cognitive processing of traumatic memories. One treatment for PTSD currently pending re-evaluation for strength of research support—based on the Tolin et al., 2015 criteria [9] adopted by the American Psychological Association's Division 12: Society of Clinical Psychology—is accelerated resolution therapy (ART), a brief therapy that was developed in 2008 and introduced in the published literature in 2012.

ART integrates aspects from various forms of psychotherapy, and uses techniques such as rapid eye movement, in vivo exposure, and image rescripting to recondition stressful memories, and reduce the physical and emotional reactions to traumatic memories [10]. The client remains in control during the sessions and does not need to talk about their traumatic experiences with the therapist to achieve recovery. It has been claimed that ART is often able to resolve a client's presenting concern in one to five sessions and that it is also beneficial for the treatment of mental health problems that are commonly associated with PTSD, such as anxiety and depression [11].

This systematic review will contribute to the literature by identifying, summarizing, and evaluating previous research that has examined the use of ART for PTSD. The aim of this review is to assess the risk of bias among studies reporting on ART and evaluate the efficacy of ART for symptoms of PTSD among adults (18 years or older) as compared to treatment as usual, pharmacotherapy, or another psychotherapy.

## Methods

This review was reported according to the Preferred Reporting Items for Systematic Review and Meta-Analysis (PRISMA) 2020 checklist [12], which can be found in S1 Appendix. The protocol for this study was published as a preprint on PsyArXiv [13], and can be viewed at the following address: https://psyarxiv.com/npqg3. This review was not registered.

### Information sources & search strategy

The search strategy was conducted in accordance with the Cochrane Handbook for Systematic Reviews of Interventions [14]. In October 2022, we consulted with a qualified research librarian to develop the search strategy and identified five electronic databases to search: MEDLINE, PsycINFO, EMBASE, CINAHL, and Scopus. The following trial registers were also searched: Cochrane Central Register of Controlled Trials (CENTRAL), US government website of clinical trials (ClinicalTrials.gov), and the WHO International Clinical Trials Registry Platform (ICTRP). Forward citation searches were conducted on applicable databases. Reference lists of relevant publications were hand-searched to identify additional publications (i.e., backward citation searches). Additional literature searches were conducted on the U.S. Department of Veterans Affairs website, https://www.va.gov/; the APA Division 12 webpage for psychological treatments of PTSD, https://div12.org/diagnosis/posttraumatic-stress-disorder/; and the official ART publications webpage, https://acceleratedresolutiontherapy.com/evidence-based/. Search terms were developed using controlled and uncontrolled vocabulary. All text searches were conducted with no limits placed on year of publication or publication status. Our search strategy was reviewed by a second research librarian using the Peer Review of Electronic Search Strategies (PRESS) checklist [15]. The initial search was completed during the fourth week of October 2022, and updated on 14 October 2023; search terms used can be found in S2 Appendix.

### Eligibility criteria

A wide net was cast given the relative paucity of studies on ART at this time. The PICOS—Population, Intervention, Comparison, Outcome, Study design—approach [16] was used to structure the eligibility criteria for this review, with the addition of 'length of follow-up'. The **population** of interest was adults 18 years of age or older—military or civilian—who met a clinical cut-off score for PTSD on a validated screening measure (e.g., PCL-5). The psychological **intervention** of interest was ART delivered in any setting (e.g., private practice, hospital, residential treatment centre, military deployment, etc.) by a qualified mental health professional with **comparison** to an active (e.g., CBT, pharmacotherapy, treatment as usual), or inactive control (e.g., wait-list) group. Intervention studies that did not include a control group (e.g., cohort studies, case series, feasibility studies) were also eligible. The primary **outcome** of interest was change in PTSD symptom severity from baseline to post intervention as measured by a validated scale of PTSD symptomatology. Secondary outcomes of interest included change in symptoms of psychological distress, depression, anxiety, and sleep quality. Additional outcomes of interest included treatment engagement and retention as measured by the number of sessions attended and the participant attrition rate prior to the end of treatment and prior to the last follow-up. Reports of harms possibly or probably associated with the intervention were also of interest. A wide range of **study designs** were considered including randomized controlled trials (RCTs), non-randomized trials, cohort studies, case-control studies, case series, and feasibility studies. Studies were excluded if they: a) did not utilize a quantitative design; b) were not primary studies or did not report on primary studies; c) were not peer-reviewed; and d) were in a language other than English with no English translation available. No restrictions were placed on **length of follow-up**.

## Screening & data extraction

Duplicates were removed and remaining citations were uploaded to Covidence for screening. Two review authors, DS and EM, independently reviewed titles and abstracts of uploaded records and discussed discrepancies until consensus was reached. Records that passed title and abstract screening underwent an independent full-text review by the same authors, with disagreements resolved through discussion until consensus was reached.

We designed a data extraction form in Covidence based on our PICOS statement which was piloted by EM on one study and independently verified by DS on a second study. Both EM and DS independently extracted data from eligible studies. Extracted data were compared and discrepancies resolved through discussion. Corresponding authors were contacted between October and November 2023 to provide further details in cases where data was missing, or important information was not reported. Where missing data could be calculated based on available data, this was done in accordance with the methods laid out in the forthcoming synthesis methods section. Based on the recommendations of Mullan, Flynn [17], details on the missing information requested, author contact, and author response are presented in S3 Appendix.

Essential data were extracted from each study including: a) author, year, and source of publication; b) characteristics of participants (e.g., age, gender, race, military or civilian status); c) number of participants enrolled and number of participants who completed treatment; d) the inclusion criteria used; e) the details of the ART intervention (e.g., number and length of sessions, who delivered them, deviations from the treatment protocol); f) the comparison used (if applicable); g) primary and secondary outcome measures; h) research design and features (e.g., sampling mechanism, treatment assignment mechanism, non-response, length of follow-up); i) treatment setting; j) adverse events; and k) the funding source. Outcomes of interest were also extracted from each study at each timepoint (i.e., pre-intervention, post-intervention, follow-up) to allow for calculation of change over time, including: a) PTSD symptom severity; b) severity of psychological distress; c) symptoms of depression; d) symptoms of anxiety; e) sleep quality; f) treatment engagement and retention as measured by the average or modal number of ART sessions attended; g) participant attrition prior to the end of treatment and prior to last follow-up; and h) reports of harms likely associated with the intervention.

## Risk of bias assessment

Risk of bias was assessed using the National Institute of Health (NIH) Study Quality Assessment Tools to evaluate methodological limitations, including sources of bias, confounding, study power, and other factors [18]. The tools are designed to facilitate critical appraisal of a study's risk of bias and culminate in a global rating of 'good', 'fair', or 'poor' quality. A 'good' study is considered to have strong internal validity and a low risk of bias. A 'fair' study is considered to have moderate internal validity along with various strengths and weaknesses, equating to a moderate risk of bias. A 'poor' study is considered to have poor internal validity and hence high risk of bias. Two review authors, EM and DS independently applied the tool to each included study and recorded supporting information and justifications for judgments of risk of bias. Discrepancies were resolved by discussion to reach consensus. The quality assessment checklists for each of the five primary studies can be found in S4 Appendix.

## Effect measures

The efficacy of ART on continuous outcome measures was compared across studies using standardized mean difference (SMD) effect sizes and their 95% confidence intervals (CIs). A small effect size was considered as 0.2–0.49, a moderate effect size as 0.5–0.79, and a large effect

size as greater than 0.8 [19]. Intention to treat (ITT) effect sizes were favoured for inclusion over completer effect sizes where a study reported both.

## Synthesis methods

Meta-analyses could not be undertaken due to heterogeneity in study designs and an insufficient number of studies with a low risk of bias. Narrative synthesis was used to describe results of individual studies and followed the synthesis without meta-analysis (SWiM) guidelines [20]. This synthesis was formatted around population features, ART intervention details, comparisons used, outcomes, study design, and treatment context. Summary of findings tables have been structured using the PICOS approach [16] within which primary studies and reports of studies are ordered by study design category (i.e., RCT vs observational) and for primary studies, quality rating.

A forest plot was generated without a pooled estimate to present primary studies grouped by whether they employed a randomized-control or observational design, following methods laid out in Chapter 12 of the Cochrane Handbook [21]. Studies were ordered within the forest plot by descending quality rating as assessed by NIH Study Quality Assessment Tools [18]. This presentation was selected to facilitate examination of heterogeneity in reported effects. Sub-group analysis could not be conducted on men versus women nor on military personnel/veterans versus civilians, due to an insufficient number of included studies. Only the effect of ART on PTSD symptom severity from pre- to post-intervention could be displayed due to a limited number of studies employing a control group, heterogeneity in the types of control groups used, limited reporting of summary statistics for secondary outcome measures, and heterogeneity in follow-up periods.

SMDs were calculated for the effect of ART on PTSD symptoms severity from pre- to post-intervention. In the case of randomized-control designs, mean gain scores, pre and post standard deviations, and pre-post correlations were extracted to calculate the SMD with 95% confidence intervals using formulae from Lipsey and Wilson [22]. In the case of non-randomized designs, dependent samples t-tests were conducted to calculate the standardized mean difference with 95% confidence intervals. Formulae from Chapter 6 of the Cochrane Handbook [23] were used to calculate the standard error, and from that the standard deviation in cases where standard deviations were not reported, but the mean difference and t-score were. A medium association, r = .30, was used when studies did not report the correlation between pre and post intervention, with sensitivity tests conducted using r = .50 and r = 10 to ensure results were not appreciably different. In one case it was necessary to combine means and standard deviations from two groups (i.e., homeless and housed veterans who were reported not to differ significantly, $p$ = .24) using the formulae $\bar{X}_{pooled} = \frac{(\bar{X}_1 * n_1) + (\bar{X}_2 * n_2)}{n_1 + n_2}$ and $s^2_{pooled} = \sqrt{\frac{(n_1 - 1)s_1^2 + (n_2 - 1)s_2^2}{n_1 + n_2 - 2}}$, respectively.

## Non-reporting bias & meta-bias assessment

Attempts were made to contact every corresponding author of the primary studies and reports of studies included in the review. Authors who responded were provided with a list of requested information. As part of this list, every author was asked if they: a) are aware of, or have conducted any unpublished research into ART, and if so to provide justification for it not being published; and b) are aware of any potential sources of bias that could have affected the conduct, interpretation, or publication of their research. All authors who responded (see S3 Appendix) stated 'no' to both questions. A narrative assessment of the risk of publication bias was conducted and instances of selective reporting within studies discussed. As outlined in

Chapter 13 of the Cochrane Handbook [24], special attention was paid to qualitative signals that raised suspicion of additional missing results. Two review authors, EM and DS independently assessed the risk of publication bias and instances of selective reporting. Disagreements were resolved through discussion to reach consensus.

Meta-bias in the current review was assessed by review authors DS and EM using the Assessing the Methodological Quality of Systematic Reviews (AMSTAR 2) checklist [25]; which can be found in S5 Appendix.

## Results

Fig 1 presents the PRISMA flow diagram depicting the article selection process and reasons for exclusion. The database and register searches resulted in the identification of 112 citations after duplicates were removed. A total of 98 citations were excluded during title and abstract screening, leaving 14 articles for full-text review. One additional article was identified through

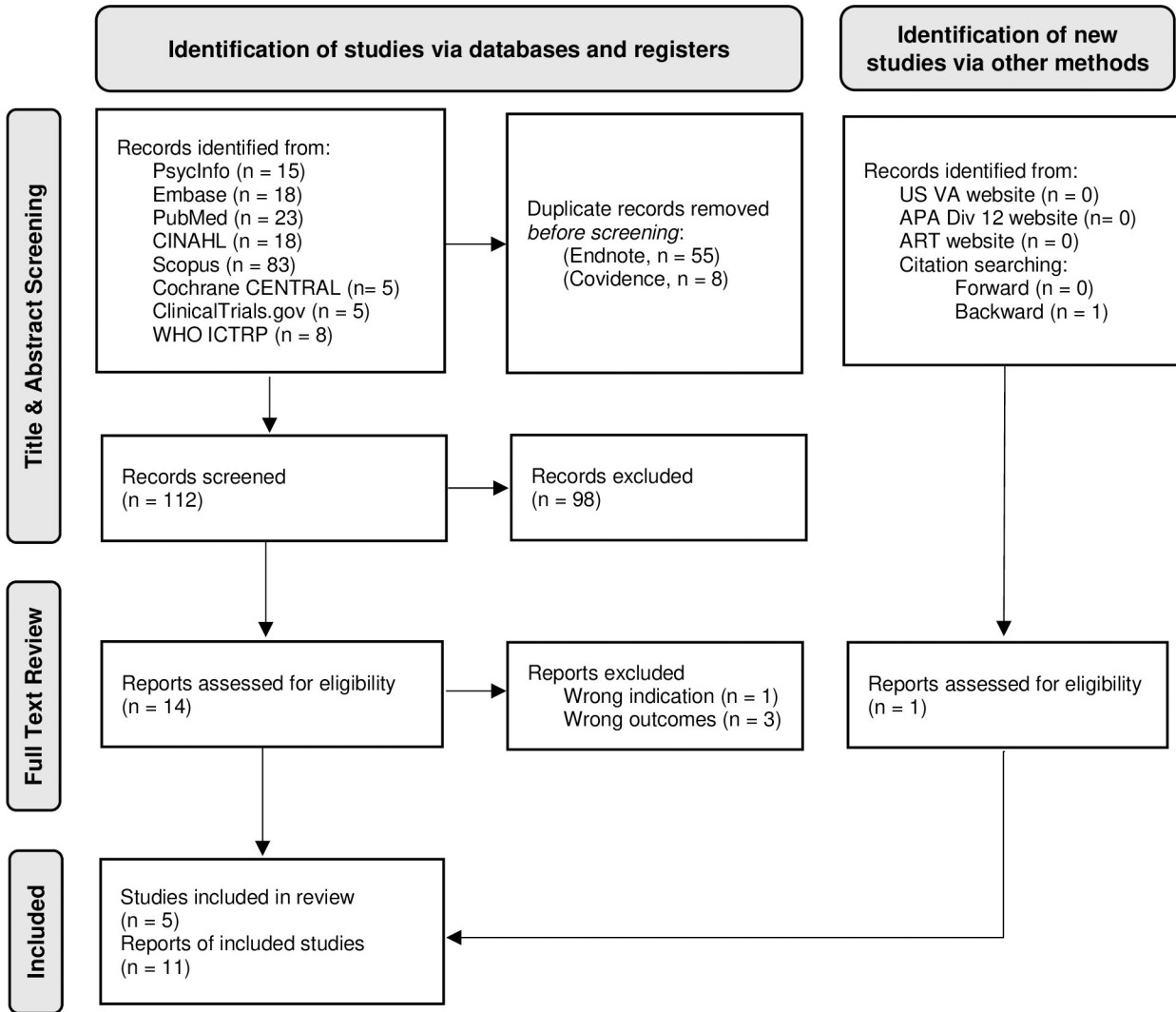

**Fig 1. PRISMA 2020 flow diagram of studies included in the review.** WHO = World Health Organization; ICTRP = International Clinical Trials Registry Platform; US VA = United States Department of Veterans Affairs; APA = American Psychological Association; ART = Accelerated Resolution Therapy.

a backward citation search and included in the full-text review. The four studies excluded during the full-text screening with reasons for exclusion, can be found in S6 Appendix. Of the 11 articles included in the review, five were primary studies and six were reports on one or more of the primary studies.

## Study characteristics

Five primary studies were included, of which two were randomized controlled trials employing a crossover design [26, 27], two were observational prospective cohort studies [28, 29], and one was a feasibility study [30]. An additional six citations were included that reported on data from the five primary studies [31–36]. The main characteristics of the included primary studies are provided in Table 1 and of the reports of studies in S1 Table, including details of the study design, population, attrition, intervention and comparison group, treatment setting, adverse events, and funding source.

Sample sizes for the included studies ranged from 6 to 140 enrolled ($N = 337$) and 5 to 89 completed ($N = 250$). The populations studied included: civilians [28], active duty military and veterans [26], homeless and housed veterans [29], female veterans who experienced military-related sexual trauma [30], and informal hospice caregivers [27]. All but one study [30] recruited males and females, though with considerable heterogeneity in the proportion of males to females. The attrition rate for the included studies ranged from 7% [27] to 36% [29].

An eligibility criterion in four of the studies required participants to have a PCL-C or PCL-M score of > 40, or a PDSQ PTSD subscale score of ≥ 5 [26, 28–30]. Buck, Cairns [27] required an Inventory of Complicated Grief (ICQ) score > 25 and a PCL-5 score > 33, rather than > 40. It should be noted that a cut-point score of 31–33 on the PCL-5 is frequently used in clinical practice, but several of the studies opted to use a higher cut-point to decrease the possible number of false positives.

The number of sessions of ART administered ranged from 1 to 5, with most participants across the included studies receiving 3–4 sessions of 60-75min in length. Session length was 45-60mins in the case of Rossiter, D'Aoust [30]. Included studies were conducted in outpatient treatment settings in the U.S. All but one study [30] included a follow-up period, the longest of which was 6 months [29]. In each study, the intervention was administered by a mental health provider qualified in ART using a manualized protocol. Three of the studies [27–29] explicitly stated that the therapist(s) used a fidelity checklist to guide the intervention; however, treatment fidelity was not reported.

Two studies included a control condition. An attention control (AC) condition was used by Kip, Rosenzweig [26], whereby participants had the option of selecting two 1-hour sessions of fitness assessment and planning or career assessment and planning. The rationale provided for the AC condition was to measure the acute effect of non-psychotherapeutic interaction with a professional while minimally withholding the amount of time to treatment crossover with ART. A waitlist (WL) control condition was employed by Buck, Cairns [27].

Two studies reported adverse events possibly or probably related to the intervention. Two participants reported feeling mildly depressed after an ART session, which resolved before subsequent sessions. There were two reports of disturbed sleep possibly related, one report of increased anxiety probably related, and one report of increased frequency of nightmares probably related to the intervention in the RCT conducted by Kip, Rosenzweig [25].

Summary statistics for each outcome measure along with a narrative summary of the key findings as reported by the author(s) of each primary study are detailed in S2 Table, and for each report of a study in S3 Table. Effect sizes and associated precision of effect were obtained from studies where reported. Within and between-groups effect sizes from pre- to post-

**Table 1. Characteristics of primary studies included in the systematic review of ART for the treatment of PTSD in adults.**

| Author (Year) | Quality Rating[a] | Study Design | Population | Attrition[b] | Inclusion Criteria | Intervention Details | Comparison Group | Treatment Setting | Adverse Events | Funding Source |
|---|---|---|---|---|---|---|---|---|---|---|
| | | | | **Randomized Controlled Trials** | | | | | | |
| **2011–2013 –Registered Clinical Trial (NCT01559688): Accelerated Resolution Therapy for Psychological Trauma ($n_{enrolled}$ = 57 active-duty military & US veterans)** | | | | | | | | | | |
| Kip, Rosenzweig, et al. (2013) | Good | RCT: prospective 2 group crossover design | Active-duty military & veterans, $n_{enrolled}{}^{c}$ = 57; $n_{completed}{}^{d}$ = 47 [$M_{Age}$ = 41, 19% female, 84% white] | 10/57 (18%) <br> 9/47 (19%) <br> 19/57 (33%) | ≥18yrs; PCL-M ≥ 40, or PDSQ PTSD subscale ≥ 5 | 2-5x ($M$ = 3.6) 60-75min sessions | Attention Control | USF College of Nursing; & NOSC, Nellis AFB | 2 possible (poor sleep); 2 probable (anxiety, nightmare) | DoD |
| **2018–2019 –Registered Clinical Trial (NCT03484338): Accelerated Resolution Therapy for Complicated Grief ($n_{enrolled}$ = 54 informal hospice caregivers)** | | | | | | | | | | |
| Buck et al. (2020) | Poor | RCT: prospective 2 group wait-listed crossover design | Informal hospice caregivers, $n_{enrolled}$ = 54, $n_{completed}$ = 43 [$M_{Age}$ = 69, 85% female, 93% white] | 4/54 (7%) <br> 7/50 (14%) <br> 11/54 (20%) | ≥60yrs; PCL-5 >33 or PDSQ PTSD subscale ≥ 5; ICQ >25 | 3-4x ($M$ = 3.7) 60-75min sessions | Waitlist | USF College of Nursing | NR | NIH |
| | | | | **Observational Trials & Studies** | | | | | | |
| **2013–2015 –Registered Clinical Trial (NCT02030522): Prospective Cohort Study of ART for the Treatment of Military Psychological Trauma ($n_{enrolled}$ = 140[†] US service members)** | | | | | | | | | | |
| Kip et al. (2016) | Fair | Observational prospective cohort | Homeless veterans, $n_{enrolled}$ = 25; $n_{completed}$ = 12 [$M_{Age}$ = 51, 9% female, 65% white]. <br><br> Non-homeless veterans, $n_{enrolled}$ = 115; $n_{completed}$ = 77 [$M_{Age}$ = 41, 6% female, 85% white] | 51/140 (36%)[e] <br> 45/89 (51%) <br> 96/140 (69%) | ≥18yrs; PCL-M ≥ 40, or PDSQ PTSD subscale ≥ 5 | 2-5x ($M$ = 3.7) 60-75min sessions | NA | HEP, Clearwater, FL; USF College of Nursing & community-based sites[f] | NR | Chris T. Sullivan Foundation |
| **2012 –Unregistered Study (Kip et al.): Brief Treatment of Symptoms of Post-Traumatic Stress Disorder by Use of Accelerated Resolution Therapy ($n_{enrolled}$ = 80 mostly civilians)** | | | | | | | | | | |
| Kip et al. (2012) | Fair | Observational prospective cohort | Adults[g], $n_{enrolled}$ = 80; $n_{completed}$ = 66 [$M_{Age}$ = 40, 77% female, 89% white] | 14/80 (18%) <br> 12/66 (18%) <br> 26/80 (33%) | 21-60yrs; PCL-C > 40, or PDSQ PTSD subscale ≥ 5 | 1-5x ($Md$ = 3) 60-75min sessions | NA | USF College of Nursing | 2 possible (mildly depressed post session) | SAMHSA; DoD |
| **2017 –Unregistered Pilot Study (Rossiter et al.): Accelerated Resolution Therapy for women veterans experiencing military sexual trauma related post-traumatic stress disorder** | | | | | | | | | | |

(*Continued*)

**Table 1.** (Continued)

| Author (Year) | Quality Rating[a] | Study Design | Population | Attrition[b] | Inclusion Criteria | Intervention Details | Comparison Group | Treatment Setting | Adverse Events | Funding Source |
|---|---|---|---|---|---|---|---|---|---|---|
| Rossiter et al. (2017) | Poor | Feasibility study | Female veterans, $n_{enrolled} = 6$, $n_{completed} = 5$ [$M_{Age} = 36$, 100% female, 80% white] | 1/6 (17%) | PCL-M $\geq$ 40 or therapist assessment of symptoms as per the Checklist for ART Standard Protocol and PDSQ PTSD subscale | 3-5x ($M = 4$) 45-60min sessions | NA | USF College of Nursing | NR | Jonas Center for Nursing; Veterans Healthcare |

*Note.* In all studies ART was administered in an individual format by one or more mental health professionals formally trained in the administration of ART.

*Abbreviations.* ART = Accelerated Resolution Therapy; PTSD = Post-Traumatic Stress Disorder; NA = Not Applicable; NR = Not Reported; USF = University of South Florida; SAMHSA = Substance Abuse and Mental Health Services Administration; DoD = Department of Defense; RCT = Randomized Controlled Trial;

Tx = Treatment; PCL = PTSD Checklist; PCL-C = PTSD Checklist Civilian; PCL-M = PTSD Checklist Military; NOSC = Naval Operational Support Center; AFB = Air Force Base; HEP = Homeless Emergency Project; Hx = History; NIH–National Institutes of Health; TATRC = Telemedicine & Advanced Technology Research Center; TBI = Traumatic Brain Injury; SOF = Special Operations Forces; ASD = Acute Stress Disorder; ICQ = Inventory of Complicated Grief.

† Information provided upon request by corresponding author.

[a] Based on the National Institutes of Health (NIH) quality rating system: good, fair, poor, cannot determine (CD), not applicable (NA), not reported (NR).

[b] Number of participants who did not complete treatment. If applicable, the second under-lined line is the number of participants lost to follow-up, and the third line is sum of treatment and follow-up attrition.

[c] $n_{enrolled}$ refers to the number of participants enrolled in an intervention, not to the number of participants screened for a study.

[d] $n_{completed}$ refers to the number of participants who completed the intervention and for which both a pre & post-intervention score was reported; however, this number does not refer to the number of participants for which 1,2, or 3-month follow-up scores were reported in applicable studies (in many cases the number of participants who completed follow-up measures is lower).

[e] The treatment non-completion rate was 48% among homeless veterans compared to 18% among veterans from community sites.

[f] Community-based sites included Orlando, Maine, and veterans Alternative.

[g] Majority were civilian; however, 6 participants in the 'civilian study' were veterans with traumas specific to military service.

intervention and pre-intervention to all reported follow-ups as reported by primary study authors are provided in Table 2 and for reports of studies in S4 Table. Effect sizes from ART and comparison groups were assigned a positive valence for outcomes favouring ART.

## Results of primary studies

Fig 2 presents a forest plot of the effect of ART on PTSD symptom severity from pre- to post-intervention among primary studies. Kip, Rosenzweig [26] found a decrease in PTSD symptom severity from pre- to post-intervention with the 95% CI ranging from a moderate to large effect (RCT; active-duty military and military veterans; low risk of bias). Buck, Cairns [27] found a substantial decrease in PTSD symptom severity from pre- to post-intervention with the 95% CI in the large effect range (RCT; informal hospice caregivers; high risk of bias). Kip, D'Aoust [29] found a substantial decrease in PTSD symptom severity from pre- to post- intervention with the 95% CI ranging from a moderate to large effect (observational; military veterans; moderate risk of bias). The results of Kip, Elk [28] were highly comparable, with the 95% CI again ranging from a moderate to large effect (observational; mostly civilians; moderate risk of bias). Rossiter, D'Aoust [30] found a very substantial decrease in PTSD symptom severity with the 95% CI in the large effect range (observational; female veteran survivors of military sexual assault; high risk of bias). The significantly higher SMD obtained from Rossiter, D'Aoust [30] as compared to the other studies may be in part attributable to the severe and sexual nature of the female veteran samples' trauma history. Perhaps female veteran survivors

**Table 2. Outcomes of primary studies included in the systematic review.**

| Author (Year) | Analyses | Follow-up | Group | Outcome Measures | Within group effect size (d)[a] [95% CI] for ART | | | Between groups effect size (d)[a] at available time points | | | |
|---|---|---|---|---|---|---|---|---|---|---|---|
| | | | | | Pre to Post | Pre to 2–3m FU | Pre to 4–6m FU | Pre Tx | Post Tx | 2–3m FU | 4–6m FU |
| **Randomized Controlled Trials** | | | | | | | | | | | |
| *2011–2013 –Registered Clinical Trial (NCT01559688): Accelerated Resolution Therapy for Psychological Trauma ($n_{enrolled}$ = 57 active-duty US military & veterans)* | | | | | | | | | | | |
| Kip, Rosenzweig, et al. (2013) | ITT | Post (ART vs AC) | ART | PCL-M (PTSD) | NP | 1.13 [NP] | | NP | 1.25 [NP] | | |
| | | | AC | BSI (psych distress) | NP | 0.82 [NP] | | NP | 0.72 [NP] | | |
| | | 3 months (ART) | | CES-D (depression) | NP | 0.85 [NP] | | NP | 1.27 [NP] | | |
| | | | | STICSA (somatic anxiety) | NP | 0.64 [NP] | | NP | 0.41 [NP] | | |
| | | | | STICSA (cognitive anxiety) | NP | 0.69 [NP] | | NP | 0.97 [NP] | | |
| | | | | PSQI (sleep) | NP | 0.51 [NP] | | NP | 0.48 [NP] | | |
| *2018–2019 –Registered Clinical Trial (NCT03484338): Accelerated Resolution Therapy for Complicated Grief ($n_{enrolled}$ = 54 informal hospice caregivers)* | | | | | | | | | | | |
| Buck et al. (2020) | Completer | Post | ART | PCL-5 (PTSD) | 2.13 [1.42,2.85] | 2.40 [1.79,3.00] | | NP | 2.11 [1.44,2.79] | NP | |
| | | 8 weeks | WL | CES-D (depression) | 1.10 [0.48,1.72] | 1.63 [1.18,2.08] | | NP | 1.09 [0.51,1.66] | NP | |
| **Observational Trials & Studies** | | | | | | | | | | | |
| *2013–2015 –Registered Clinical Trial (NCT02030522): Prospective Cohort Study of Accelerated Resolution Therapy for Treatment of Military Psychological Trauma ($n_{enrolled}$ = 140[†] US service mbrs)* | | | | | | | | | | | |
| Kip et al. (2016) | Completer[b] | Post | ART | ***Homeless Veterans*** | | | | | | | |
| | | 6 months | | PCL-M (PTSD) | NP | | NP | NP | 0.59 [NP] | | 0.59 [NP] |
| | | | | BSI (psych distress) | NP | | | NP | 0.12 [NP] | | |
| | | | | CES-D (depression) | NP | | | NP | 0.35 [NP] | | |
| | | | | STICSA (state anxiety) | NP | | | NP | 0.65 [NP] | | |
| | | | | PSQI (sleep) | NP | | | NP | 0.66 [NP] | | |
| | | | | ***Housed Veterans*** | | | | | | | |
| | | | | PCL-M (PTSD) | NP | | NP | | | | |
| | | | | BSI (psych distress) | NP | | | | | | |
| | | | | CES-D (depression) | NP | | | | | | |
| | | | | STICSA (state anxiety) | NP | | | | | | |
| | | | | PSQI (sleep) | NP | | | | | | |
| *2012 –Unregistered Study (Kip et al.): Brief Treatment of Symptoms of Post-Traumatic Stress Disorder by Use of Accelerated Resolution Therapy ($n_{enrolled}$ = 80 mostly civilians)* | | | | | | | | | | | |
| Kip et al. (2012) | Completer[b] | Post | ART | PCL-C (PTSD) | 1.72 [NP] | 1.98 [NP] | NP | | | | |
| | | 2 months | | BSI (psych distress) | 1.74 [NP] | 1.57 [NP] | NP | | | | |
| | | | | CES-D (depression) | 1.41 [NP] | 1.46 [NP] | NP | | | | |
| | | | | STICSA (somatic anxiety) | 1.11 [NP] | 1.10 [NP] | NP | | | | |
| | | | | STICSA (cognitive anxiety) | 1.62 [NP] | 1.14 [NP] | NP | | | | |
| | | | | PSQI (sleep) | 0.87 [NP] | 0.70 [NP] | NP | | | | |

*(Continued)*

**Table 2.** (*Continued*)

| Author (Year) | Analyses | Follow-up | Group | Outcome Measures | Within group effect size ($d$)[a] [95% CI] for ART | | | | Between groups effect size ($d$)[a] at available time points | | | |
|---|---|---|---|---|---|---|---|---|---|---|---|---|
| | | | | | Pre to Post | Pre to 2–3m FU | Pre to 4–6m FU | Pre Tx | Post Tx | 2–3m FU | 4–6m FU |
| **2017 –Unregistered Pilot Study (Rossiter et al.): Accelerated Resolution Therapy for women veterans experiencing military sexual trauma related post-traumatic stress disorder** | | | | | | | | | | | | |
| Rossiter et al. (2017) | Completer | Post | ART | PCL-M (PTSD) | 2.32 [NP] | | | | | | |
| | | | | BSI (psych distress) | 1.28 [NP] | | | | | | |
| | | | | CES-D (depression) | 0.47 [NP] | | | | | | |
| | | | | STICSA (anxiety) | 0.78 [NP] | | | | | | |
| | | | | PSQI (sleep) | 0.35 [NP] | | | | | | |

*Note.* Empty cells indicate no data collected for that time point. Within and between effect sizes and 95% CI's are as reported in each study. Positive effect sizes reflect an improvement in symptoms relative to baseline or control (or in the cases of Kip et al., 2016 –homeless relative to housed veterans); negative effect sizes reflect a degradation in symptoms relative to baseline or control.

*Abbreviations.* NP = Not Provided in publication & not available upon request; LP = Low Precision (e.g., boxplot provided, but no precise values); ART = Accelerated Resolution Therapy; PTSD = Post-Traumatic Stress Disorder; FU = Follow-up; ITT = Intention To Treat; PCL = PTSD Checklist; BSI = Brief Symptom Inventory; CES-D = Center for Epidemiological Studies Depression Scale; STICSA = State-Trait Inventory for Cognitive and Somatic Anxiety; PSQI = Pittsburgh Sleep Quality Index; AC = Attention Control; Vets = Veterans; CST = Civilian Sexual Trauma; MST = Military Sexual Trauma; TBI = Traumatic Brain Injury; SOF = Special Operations Forces; ASD = Acute Stress Disorder; Dx = Diagnosis.

† Information provided upon request by corresponding author.

[a] According to Cohen (1988), $d < 0.2$ indicates no difference, $d = 0.2$–$0.49$ indicates a small difference, $d = 0.5$–$0.79$ indicates a medium difference, and $d \geq 0.8$ indicates a large difference.

[b] Conducted completer analysis; however, also conducted a sensitivity analysis comparing completers to drop-outs on demographic and baseline diagnostic variables.

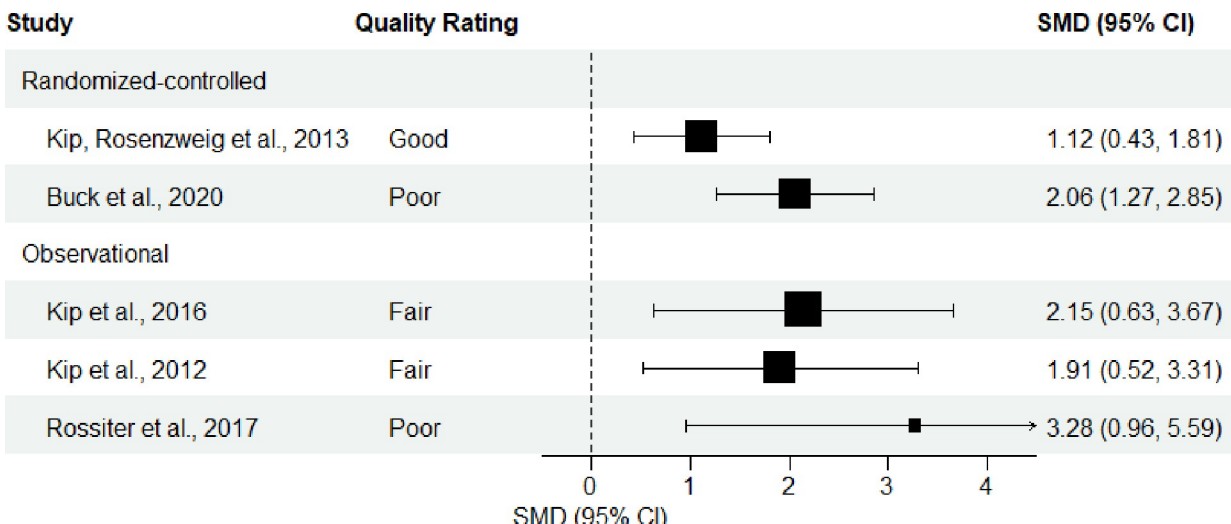

**Fig 2. Forest plot of effect of ART on PTSD severity from pre- to post-intervention.** *Note.* Quality rating is as assessed by the NIH Study Quality Assessment Tools (National Institutes of Health, 2021). The size of the point estimate boxes is scaled to log10 (N), where N = number of participants in each study who completed treatment (i.e., a larger box indicates a larger sample). The Standardized Mean Difference (SMD) is a measure of effect size and was calculated by the review authors using summary data (and therefore may differ slightly from those reported by study authors as found in Table 4). The SMDs were calculated assuming a medium association of r = .30 between pre and post intervention. Sensitivity tests were conducted with low and high associations of r = .10 and r = .50, respectively, and did not differ significantly. According to Cohen (1992), a small effect size is considered 0.2–0.49, a moderate effect size is 0.5–0.79, and a large effect size is 0.80 or greater. Note that a given effect size is not on its own indicative of whether a given effect is or is not clinically significant.

of military sexual assault benefit more than other populations from ART. However, given the high risk of bias, and the very small sample size, a larger and better-quality study would need to be conducted to test this hypothesis.

## Results of secondary studies

The six secondary studies reported significant reductions on the PCL (C or M) from pre- to post-intervention within sub-groups of the primary studies. For example, Kip, Sullivan [31] reported that a sub-group of participants who screened for PTSD and comorbid depression had more pronounced and sustained treatment response than those who screened for PTSD only, with $d = 2.37$ at post-treatment, and $d = 3.01$ at 4-month follow-up. In a comparison of homeless and housed veterans, Kip, D'Aoust [29] reported a moderate effect in favor of a stronger treatment response among homeless veterans, $d = 0.59$; however, this result did not reach statistical significance, $p = .14$. In a comparison of participants with high vs low PTSD severity, Witt [32] reported a small effect in favor of a stronger treatment response among those with higher severity of PTSD symptomatology; however, levels of traumatic guilt, depression, and anxiety were not significant predictors of PTSD symptom reduction. In a comparison of participants based on previous treatment history for PTSD, Pang, Murn [33] reported that participants who received another 1st line psychotherapy experienced the largest reduction in symptoms, $d = 1.88$, followed by treatment naïve participants, $d = 1.48$; participants who had received pharmacotherapy only, $d = 1.11$, or another psychotherapy, $d = 1.03$, also experienced significant reductions in symptoms, but to a lesser degree. In a comparison of treatment response between civilian and military participants stratified by gender and history of sexual trauma, Kip, Hernandez [35] reported a moderate effect in favor of stronger treatment response amongst civilians, $d = 0.51$, though after adjusting for history of head trauma and sleep quality the difference was no longer significant. Hardwick [34] reported that a civilian sample had a significantly stronger response to treatment as compared to a military sample post-intervention, $d = 1.77$; however, at three month follow up the differential response was considerably smaller, $d = 0.29$. Likewise, Kip, Berumen [36], found no significant difference in response to treatment in U.S. service members and veterans across different severities of traumatic brain injury (TBI).

## Non-reporting bias & meta-bias

No missing studies for the synthesis of the effect of ART on PTSD symptoms severity from pre- to post-intervention were detected; however, there were qualitative indications that some results collected were not reported in the published literature. For example, Kip, Elk [28] collected 4-month follow-up data but did not report on it. Also, many authors did not report 95% CIs for measures of effect and were unable to provide this information upon request.

Meta-bias in the current review was assessed to be low. All applicable items in the AMSTAR 2 checklist [25], received a yes. Only one item—regarding the use of a protocol—received a partial yes, as the protocol, though published as a pre-print, was not registered. Based on recommendations from Uttley and Montgomery [37], we have provided additional context pertinent to assessing meta-bias: a) this review was not sponsored by a public, private, or non-profit entity; b) none of the reviewers were trained in ART; and c) reviewers have no financial or research interest in the outcome of this review.

## Discussion

Based on the Tolin et al., 2015 criteria [9] adopted by the American Psychological Association's Division 12: Society of Clinical Psychology, evidence for a treatment based on a systematic

review of published meta-analyses of RCTs can be rated as *very strong, strong, weak, or having insufficient evidence*. Currently ART is pending re-evaluation based on these criteria, and this systematic review will not change that for three reasons: 1) it is the first and only systematic review of ART to date, 2) a quantitative synthesis (i.e., meta-analyses) was not possible, and 3) for the Tolin criteria to be assessed a review of multiple quantitative reviews is needed. Unfortunately due to the small number of high-quality studies on ART at this time, there is an insufficient research base to be able to assess ART against the Tolin criteria. However, this systematic review lays the groundwork for future research into the use of ART for the treatment of PTSD. It may also prompt an upgrading of the strength of research support for treating PTSD from 'modest' to 'strong' based on the older Chambless et al., 1998 criteria [38]. In this review multiple well-designed studies (i.e., fair to good quality) conducted by independent investigators have converged to support the efficacy of ART for improving symptoms of PTSD, as well as symptoms of depression, mental distress, anxiety, and sleep disturbance from pre- to post-intervention and follow-up, with little risk of harm, and a reasonable investment of resources.

## Limitations and future directions

Results from this systematic review should be considered in the light of several limitations. First, this review included a small number of studies. Only five primary studies were included in this review, of which only two were RCTs. These studies had relatively small sample sizes ranging from 6 to 140 participants. Second, only two of the studies were comparative, and both used different comparisons, which meant that the original review question to compare ART to TAU, pharmacotherapy, or another therapy, could not be addressed in the synthesis. As a result, the effect of ART could only be assessed from pre- to post- intervention. This allows for some conclusions to be drawn regarding whether ART improves symptoms associated with PTSD; but does not allow for conclusions to be drawn regarding how ART compares to other treatments. Third, there was heterogeneity in the sub-groups considered across studies, which limited the ability to draw reliable conclusions about the efficacy of ART for treating PTSD in specific sub-populations within the synthesis. Fourth, several included studies did not report complete summary statistics, and authors were unable to provide them upon request. As such, it was not possible to calculate SMDs for all outcomes of interest and for all timepoints of interest. Fifth, the quality of included studies ranged from 'poor' to 'good,' with only one study being rated as 'good,' suggesting reasonable potential for risk of bias.

ART involves the combination of multiple different forms of psychotherapy, such as EMDR, in vivo exposure, and image rescripting. It is not currently known to what degree each component contributes to the outcomes found. A dismantling study is needed to determine which components of ART may be resulting in the reported reduction in PTSD symptoms to determine the most effective intervention strategies, and to assess whether ART's propriety protocol provides appreciable value added over other approaches. Additional low risk of bias comparative studies examining the efficacy of ART for PTSD are needed to allow for statistical synthesis.

## Conclusion

This was the first systematic review on the efficacy of ART for reducing symptoms of PTSD. All included studies reported a significant reduction in symptoms of PTSD from pre- to post-intervention as well as significant reductions in symptoms of depression, mental distress, anxiety, and sleep dysfunction from pre- to post-intervention. ART shows some promise as a time-efficient treatment for symptoms of PTSD in adults and may merit an upgrade on the strength

of research support from 'modest' to 'strong' based on the Chambless et al., 1998 criteria [38]; however, more high-quality research is needed, including RCTs that compare ART to current 1st line psychotherapies for PTSD, before stronger conclusions based on the Tolin et al., 2015 criteria [9] can be drawn.

## Supporting information

**S1 Appendix. PRISMA 2020 checklist.**
(PDF)

**S2 Appendix. Search terms.**
(PDF)

**S3 Appendix. Missing information requested, author contact details, and response rate from authors contacted Oct/Noc 2023.**
(PDF)

**S4 Appendix. NIH study quality assessment checklists.**
(PDF)

**S5 Appendix. Assessing the Methodological Quality of Systematic Reviews (AMSTAR 2) checklist.**
(PDF)

**S6 Appendix. Reasons for excluding studies or reports of studies during full text screening.**
(PDF)

**S1 Table. Characteristics of reports of studies included in the systematic review of ART for the treatment of PTSD in adults.**
(PDF)

**S2 Table. Summary statistics and key findings reported by authors of primary studies included in the systematic review.**
(PDF)

**S3 Table. Summary statistics and key findings reported by authors of reports of studies included in the systematic review.**
(PDF)

**S4 Table. Outcomes of reports of studies included in the systematic review.**
(PDF)

## Acknowledgments

We would like to thank Alison Farrell of the Memorial University Health Sciences Library who aided the team with the literature search strategy and selection of search terms. We would also like to thank the study authors who took the time to respond to our requests for information.

## Author Contributions

**Conceptualization:** David Paul Storey, Emily Claire Shaw Marriott.

**Formal analysis:** David Paul Storey, Emily Claire Shaw Marriott.

**Investigation:** David Paul Storey, Emily Claire Shaw Marriott.

**Methodology:** David Paul Storey, Emily Claire Shaw Marriott.

**Supervision:** Joshua A. Rash.

**Validation:** Joshua A. Rash.

**Visualization:** David Paul Storey, Emily Claire Shaw Marriott.

**Writing – original draft:** David Paul Storey, Emily Claire Shaw Marriott.

**Writing – review & editing:** David Paul Storey, Emily Claire Shaw Marriott, Joshua A. Rash.

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
