## [Decision Letter · Decision Letter 0]

16 Jul 2024

PMEN-D-24-00106

Accelerated Resolution Therapy (ART) for the treatment of posttraumatic stress disorder in adults: A systematic review

PLOS Mental Health

Dear Dr. Storey,

Thank you for submitting your manuscript to PLOS Mental Health. After careful consideration, we feel that it has merit but does not fully meet PLOS Mental Health’s publication criteria as it currently stands. Therefore, we invite you to submit a revised version of the manuscript that addresses the points raised during the review process.

We look forward to receiving your revised manuscript.

Kind regards,

Hongru Zhu

Academic Editor

PLOS Mental Health

Journal Requirements:

Additional Editor Comments (if provided):

Reviewers' comments:

Reviewer's Responses to Questions

**Comments to the Author**

1. Does this manuscript meet PLOS Mental Health’s publication criteria? Is the manuscript technically sound, and do the data support the conclusions? The manuscript must describe methodologically and ethically rigorous research with conclusions that are appropriately drawn based on the data presented.

Reviewer #1: Yes

Reviewer #2: Yes

2. Has the statistical analysis been performed appropriately and rigorously?

Reviewer #1: Yes

Reviewer #2: Yes

3. Have the authors made all data underlying the findings in their manuscript fully available (please refer to the Data Availability Statement at the start of the manuscript PDF file)?

Reviewer #1: Yes

Reviewer #2: Yes

4. Is the manuscript presented in an intelligible fashion and written in standard English?

Reviewer #1: Yes

Reviewer #2: Yes

5. Review Comments to the Author

Reviewer #1: Intro: Although stats can vary, what is the prevalence among the gen pop, civilians?

Provide a brief description of the common forms of therapy to differentiate between those and ART

Can you provide sample search terms?

What was your PICO statement?

Elaborate on forest plot and why this is used.

Not sure who the PCL-5 score of 33 is less stringent when 33 is the typical cutoff

You have a lot of room to expand in the discussion section-why is ART important and why should additional studies be implemented for this population? Make a strong point here.

Reviewer #2: Journal: Plos mental health

In the manuscript titled “Accelerated Resolution Therapy (ART) for the treatment of posttraumatic stress disorder in adults: A systematic study”, David et al firstly reported the efficacy of ART in the treatment of PTSD with a well-confirmed meta-analysis method. The authors found that there was a significant reduced of ART in PTSD symptoms from pre-post intervention. Although this is a well-established meta-analysis confirmed strictly in accordance with the guideline of Cochrane systematic reviews, some issues and some missed information need to be issued.

1. The details about publications which was included or excluded from this study need to be supplied, and a study flowchart need to be added in the revised manuscript.

2. Since only 11 related publications were included in the analysis, other potential database should be considered to add in screening the papers.

3. The tables, including table 1-2 lost some information due to the wrongly setting of output form.

6. PLOS authors have the option to publish the peer review history of their article (what does this mean?). If published, this will include your full peer review and any attached files.

**Do you want your identity to be public for this peer review?** For information about this choice, including consent withdrawal, please see our Privacy Policy.

Reviewer #1: No

Reviewer #2: No

---

## [Decision Letter · Decision Letter 1]

15 Aug 2024

Accelerated Resolution Therapy (ART) for the treatment of posttraumatic stress disorder in adults: A systematic review

PMEN-D-24-00106R1

Dear Mr. Storey,

We are pleased to inform you that your manuscript 'Accelerated Resolution Therapy (ART) for the treatment of posttraumatic stress disorder in adults: A systematic review' has been provisionally accepted for publication in PLOS Mental Health.

Best regards,

Hongru Zhu

Academic Editor

PLOS Mental Health

Reviewer Comments (if any, and for reference):

Reviewer's Responses to Questions

**Comments to the Author**

1. If the authors have adequately addressed your comments raised in a previous round of review and you feel that this manuscript is now acceptable for publication, you may indicate that here to bypass the “Comments to the Author” section, enter your conflict of interest statement in the “Confidential to Editor” section, and submit your "Accept" recommendation.

Reviewer #1: All comments have been addressed

Reviewer #2: All comments have been addressed

2. Does this manuscript meet PLOS Mental Health’s publication criteria? Is the manuscript technically sound, and do the data support the conclusions? The manuscript must describe methodologically and ethically rigorous research with conclusions that are appropriately drawn based on the data presented.

Reviewer #1: Yes

Reviewer #2: Yes

3. Has the statistical analysis been performed appropriately and rigorously?

Reviewer #1: Yes

Reviewer #2: Yes

4. Have the authors made all data underlying the findings in their manuscript fully available (please refer to the Data Availability Statement at the start of the manuscript PDF file)?

Reviewer #1: Yes

Reviewer #2: Yes

5. Is the manuscript presented in an intelligible fashion and written in standard English?

Reviewer #1: Yes

Reviewer #2: Yes

6. Review Comments to the Author

Reviewer #1: Thank you for adding the prevalence and PICO. It really helps set the tone for the rest of your paper. Great work!

Reviewer #2: After the manuscript was reverted, this paper is nearly ready to be accepted for this journal; only one note needs to be addressed: the authors supplied the revised manuscript but need a clear copy. The latter makes the reviewer easily read your paper.

7. PLOS authors have the option to publish the peer review history of their article (what does this mean?). If published, this will include your full peer review and any attached files.

**Do you want your identity to be public for this peer review?** For information about this choice, including consent withdrawal, please see our Privacy Policy.

Reviewer #1: No

Reviewer #2: No
